# Female resistance to pneumonia identifies lung macrophage nitric oxide synthase-3 as a therapeutic target

Zhiping Yang[1], Yuh-Chin T Huang[2], Henry Koziel[3], Rini de Crom[4], Hartmut Ruetten[5], Paulus Wohlfart[5], Reimar W Thomsen[6], Johnny A Kahlert[6], Henrik Toft Sørensen[6], Szczepan Jozefowski[7], Amy Colby[1], Lester Kobzik[1,8]*

[1]Department of Environmental Health, Harvard School of Public Health, Boston, United States; [2]Human Studies Division, National Health and Environmental Effects Research Laboratory, US Environmental Protection Agency, Chapel Hill, United States; [3]Division of Pulmonary, Critical Care and Sleep Medicine, Beth Israel Deaconness Medical Center, Boston, United States; [4]Department of Cell Biology and Genetics, Erasmus University Medical Center, Rotterdam, Netherlands; [5]Diabetes Division, Sanofi Research and Development, Frankfurt, Germany; [6]Department of Clinical Epidemiology, Aarhus University Hospital, Aarhus, Denmark; [7]Department of Immunology, Jagiellonian University Medical College, Kraków, Poland; [8]Department of Pathology, Brigham and Women's Hospital, Boston, United States

**Abstract** To identify new approaches to enhance innate immunity to bacterial pneumonia, we investigated the natural experiment of gender differences in resistance to infections. Female and estrogen-treated male mice show greater resistance to pneumococcal pneumonia, seen as greater bacterial clearance, diminished lung inflammation, and better survival. In vitro, lung macrophages from female mice and humans show better killing of ingested bacteria. Inhibitors and genetically altered mice identify a critical role for estrogen-mediated activation of lung macrophage nitric oxide synthase-3 (NOS3). Epidemiologic data show decreased hospitalization for pneumonia in women receiving estrogen or statins (known to activate NOS3). Pharmacologic targeting of NOS3 with statins or another small-molecule compound (AVE3085) enhanced macrophage bacterial killing, improved bacterial clearance, and increased host survival in both primary and secondary (post-influenza) pneumonia. The data identify a novel mechanism for host defense via NOS3 and suggest a potential therapeutic strategy to reduce secondary bacterial pneumonia after influenza.

*For correspondence: lkobzik@hsph.harvard.edu

## Introduction

Bacterial pneumonia remains a major cause of morbidity and mortality (*Mizgerd, 2006*; *Shrestha et al., 2013*). One approach to the problem might be to enhance innate immunity to infection. Normal host defenses are already quite robust, albeit imperfect, as they keep the incidence of pneumonia much lower than possible given the normal nocturnal aspiration of nasopharyngeal bacteria, example *Streptococcus pneumoniae* (*Gleeson et al., 1997*; *Dockrell et al., 2012*; *Donkor, 2013*). The resident alveolar macrophage (AM) functions as a 'first responder' phagocyte, ingesting and killing inhaled bacteria (*Green and Kass, 1964*; *Fels and Cohn, 1985*; *Hussell and Bell, 2014*). The importance of this function of AMs is indicated by greater susceptibility to infection and

**eLife digest** Pneumonia is a disease that is commonly caused by a bacterial infection and results in the lungs becoming inflamed. Pneumonia is a serious condition and can lead to hospitalization and sometimes death. However, women—and other female animals—are less likely than males to get pneumonia and are more likely to survive if they do. Understanding this sex-based difference may help to develop treatments or preventive actions that either reduce the number of people who get pneumonia or help infected patients to recover.

Bacteria from the nose—including those that cause pneumonia—frequently enter the lungs during sleep. Luckily, the body has very robust defense mechanisms against such invasions; the immune system immediately deploys cells called macrophages as a 'first response' to devour and kill invading bacteria in the lungs. However, this system is not perfect, particularly if an individual has a weakened immune system or if they are already suffering with a respiratory infection. Indeed, many individuals with severe influenza infections are hospitalized as a result of pneumonia.

Yang et al. studied why females are more able to fend off pneumonia and found that estrogen, the main female sex hormone, boosts the ability of the macrophages to kill bacteria. Treating male mice with estrogen also boosted their immune system's ability to kill off bacteria in the lungs.

Investigating further, Yang et al. found that the estrogen worked by increasing the number of proteins produced from one gene called *NOS3*. Female mice lacking NOS3 proteins lost their pneumonia-fighting advantage. A widely used class of drugs called statins, which are used to treat cardiovascular disease, boosts the activity of the NOS3 gene. Yang et al. therefore wondered whether treatment with either estrogen or statins might prevent pneumonia, or help patients with pneumonia fight off the infection.

Using a large database of information about healthcare in Denmark, Yang et al. assessed the relationship between taking these drugs and the risk of pneumonia. When several confounding factors (such as unrelated diseases that the patient was suffering from) are taken into account, the data show that the women were less likely to be hospitalized for pneumonia if they were taking statins or estrogens. Those taking both treatments had an even lower risk.

Yang et al. also found that treating mice with statins or an experimental drug that boosts NOS3 activity increased the ability of the animals to fight off pneumonia-causing bacteria—even if they also had influenza—and increased the likelihood that mice already infected with pneumonia would survive. Further studies will be needed to determine if statins or the experimental drug might also help to prevent pneumonia in human patients with influenza.

diminished bacterial clearance after their experimental depletion (*Dockrell et al., 2003*; *Ghoneim and McCullers, 2013*) or their impairment by clinical risk factors such as recent influenza infection (*Sun and Metzger, 2008*).

To guide investigation of possible targets to improve or restore lung macrophage antibacterial function, we sought to exploit the natural experiment of gender differences in resistance to infections. Experimental models find that female mice show greater systemic resistance to pneumococci (*Weiss et al., 1973*; *Kadioglu et al., 2011*) and to many other (but not all) pathogens (*McClelland and Smith, 2011*). Epidemiologic studies of human pneumonia observe a greater incidence of community-acquired pneumonia in males (*Gutiérrez et al., 2006*) and show that males are at greater risk than females for pneumonia after admission to hospital after adjusting for other risk factors such as smoking and alcohol use (*Offner et al., 1999*; *Andermahr et al., 2002*; *Gannon et al., 2004*).

We chose to address this problem by using a model of pneumococcal infection that approximates the frequent challenge to lung defenses by small numbers of bacteria and comparing responses in male and female mice. We identified greater female resistance to infection, mediated in large part by estrogen-dependent activation of constitutive AM nitric oxide synthase-3 (NOS3). Pharmacologic agents that enhance NOS3 function improved resistance in mouse models of both primary lung infection and post-influenza secondary pneumonia, suggesting a strategy to enhance resistance to common and serious lung infections.

## Results

### Superior resistance of female mice to pneumococcal pneumonia

We tested effects of a relatively small bacteria inoculum size to simulate the common, low-level challenge to the lungs from aspiration of upper airway bacteria. Female mice and estrogen-treated male mice showed greater clearance of bacteria from the lungs and less acute inflammation (neutrophil influx) compared to normal or sham-treated males 24 hr after inoculation of *S. pneumoniae* (*Figure 1A,B*). Pilot experiments compared efficiency of delivery in male vs female mice by measuring bacterial CFUs 5 min after instillation. The results showed slightly greater initial bacterial loads in female mice ($p < 0.02$, n = 12/group), indicating that the female advantage does not reflect a lower inoculum due to anatomic

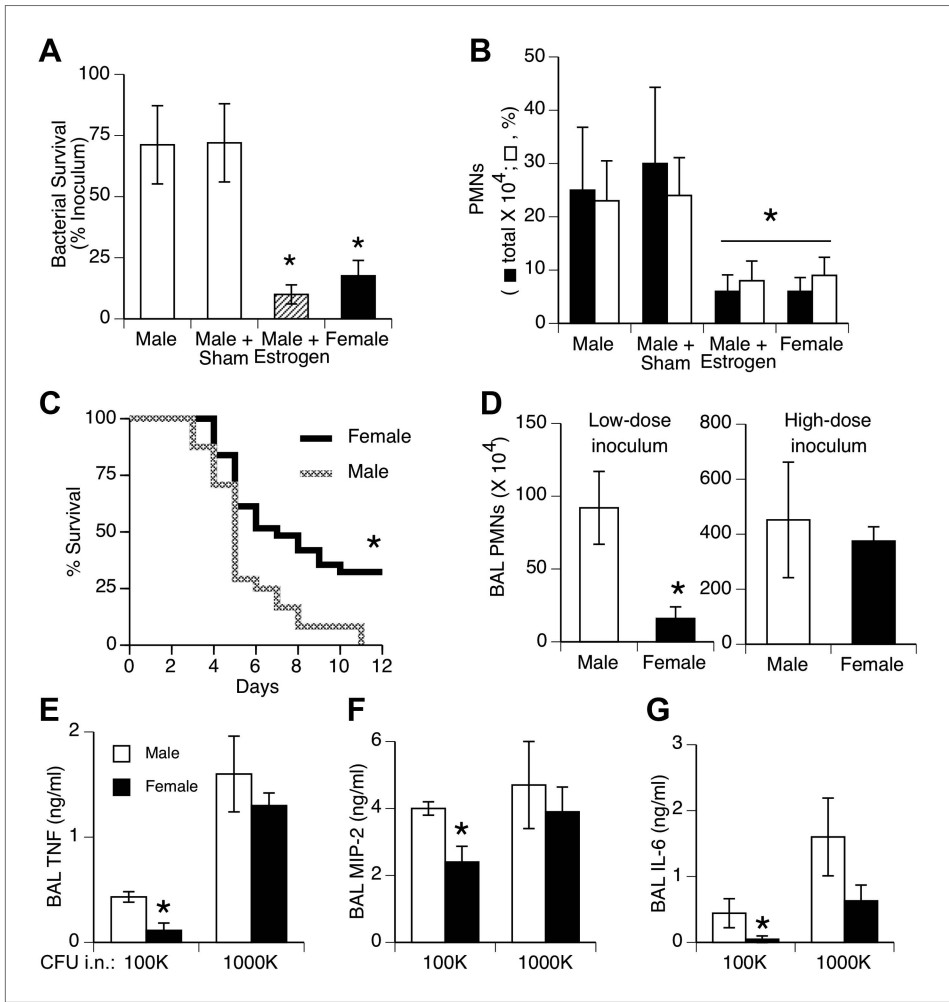

**Figure 1**. Females show greater resistance to pneumococcal pneumonia. (**A**) Twenty-four hours after intranasal (i.n.) inoculation of *S. pneumoniae* (~$10^5$ CFU), lung samples from female mice (and estrogen-treated male mice via subcutaneous slow-release 17-beta-estradiol pellets, ~70 µg/day) contain fewer live bacteria than seen in male mice (n $\geq$ 12, * = $p < 0.01$ vs control or sham-treated males) and (**B**) show less acute inflammation (BAL neutrophils, n $\geq$ 12, * = $p < 0.01$). (**C**) After i.n. pneumococcus, female mice show significantly greater survival than male mice (2.5 × $10^5$ CFU, n $\geq$ 24, * = $p < 0.01$). Gender differences in pneumonic inflammation are seen with low (4 × $10^5$ CFU), but not high (11 × $10^5$), bacterial inocula, measured as BAL neutrophilia (**D**) or BAL cytokines TNF (**E**), MIP-2 (**F**), or IL-6 (**G**), (n $\geq$ 3, * = $p < 0.05$).
The following figure supplement is available for figure 1:

**Figure supplement 1**. Respiratory burst by male and female alveolar macrophages.

or size differences. Greater female resistance was also observed in longer duration survival studies (*Figure 1C*). The gender differences we observed with $10^5$ colony-forming units (CFU) were not seen if a lethal inoculum (~11-fold higher) of bacteria was used, as both genders showed markedly increased inflammation and lung cytokine levels (*Figure 1D–G*).

## Superior bacterial killing by female alveolar macrophages

To compare the innate antibacterial function of alveolar macrophages from both genders, we measured phagocytosis and killing of pneumococci in vitro by AMs from normal mice and humans. Analysis of bacterial binding and internalization showed no differences between male and female murine AMs (*Figure 2A,B*; similar data with human AMs not shown). In contrast, killing of internalized bacteria was greater in female AMs than male AMs in mouse samples challenged with *S. pneumoniae*, as well as with other lung pathogens *Staphylococcus aureus*, and *Escherichia Coli* (*Figure 2C–E*). Similarly, normal human female AMs showed greater killing of ingested pneumococci than their male counterparts (*Figure 2F*).

## NOS3 mediates the female advantage

Mechanisms for killing phagocytosed bacteria include production of reactive oxygen species (ROS), example, superoxide produced by NADPH oxidase, and reactive nitrogen intermediates (RNI), example nitric oxide produced by nitric oxide synthases (NOS). We first used mice genetically deficient in NADPH oxidase (*Morgenstern et al., 1997*) to more directly evaluate the importance of this pathway in vivo. Bacterial clearance in NADPH oxidase-null mice was significantly impaired, but did not eliminate the gender difference, since both male and female mice were affected approximately equally (*Figure 3A*). Additional experiments showed no difference in generation of ROS by male and female AMs stimulated to undergo a respiratory burst in vitro (*Figure 1—figure supplement 1*). Similarly, quantitative real-time PCR using primers for NADPH oxidase components (phox22, 47, 91) or myeloperoxidase showed no difference in expression (data not shown).

To explore the potential role of RNI, we tested the effect of NOS inhibitors on bacterial killing by female murine AMs in vitro. As shown in *Figure 3B*, the non-selective NOS blocker nitro-l-arginine (NLA) caused substantial inhibition of female AM killing, while its stereoisomer controls NDA and the inducible NOS (iNOS or NOS2)-specific inhibitor 1400W had no effect. This suggested a role for

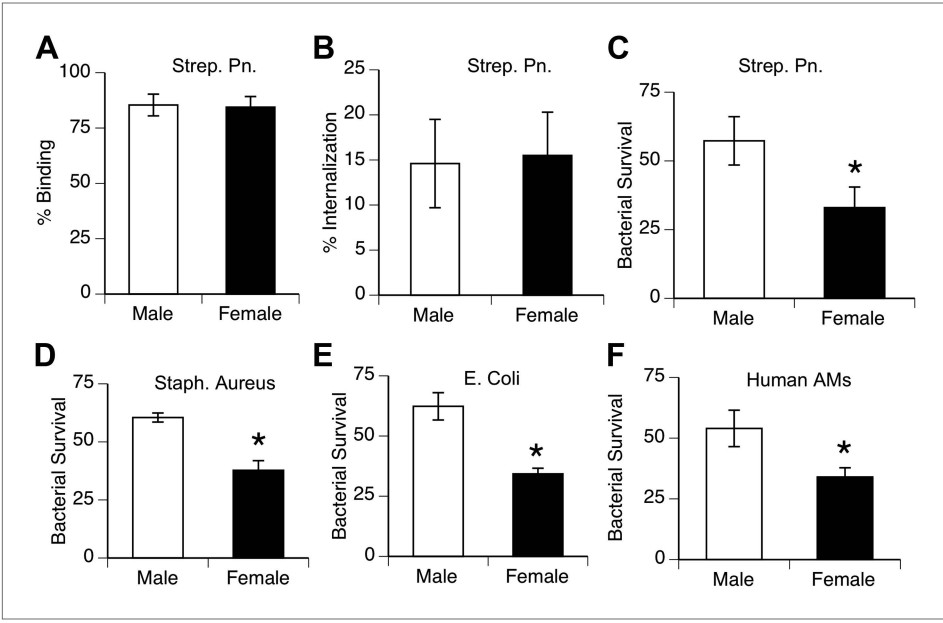

**Figure 2**. Female alveolar macrophages show better killing of ingested bacteria. Binding (**A**) and internalization (**B**) of *S. pneumoniae* in normal male and female AMs is similar. Female AMs kill more internalized bacteria than male AMs in assays using pneumococci (**C**) (n $\geq$ 11, * = p < 0.01), *S. aureus* (**D**) or *E. coli* (**E**), (n $\geq$ 3, * = p < 0.01). (**F**). Normal human female AMs also show greater killing of internalized pneumococci, (n $\geq$ 5, * = p < 0.01).

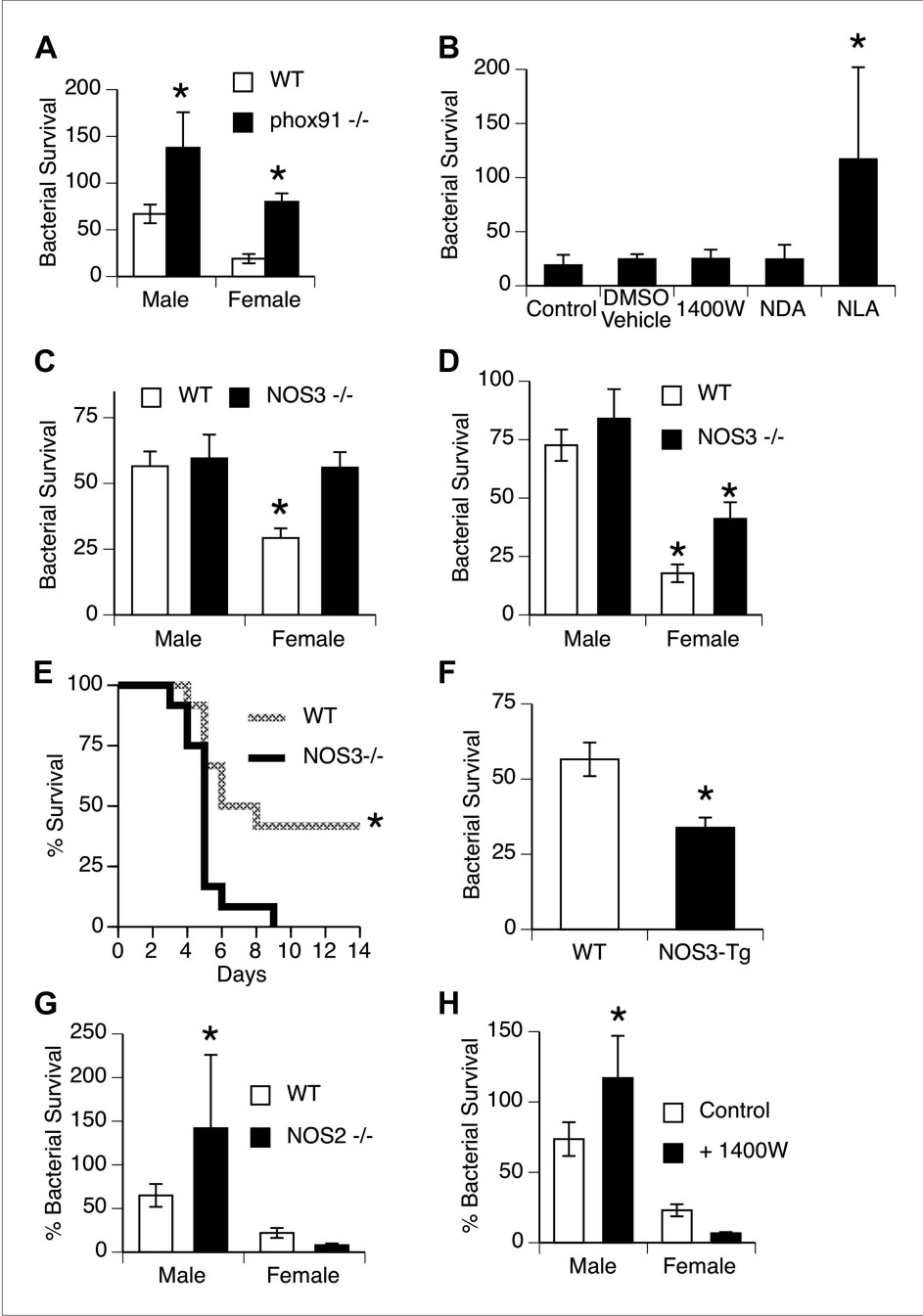

**Figure 3**. NOS3 and female resistance to pneumococcal pneumonia. (**A**) NADPH oxidase deficient (phox91$^{-/-}$) mice show comparable reduction in bacterial clearance in both male and female mice (n = 6, * = p < 0.01 vs wild-type). (**B**) In vitro killing of pneumococci by normal mouse female AMs is inhibited by the non-selective NOS inhibitor nitro-L-arginine (NLA), but not by its inactive stereo-isomer, nitro-D-arginine (NDA), nor by the type 2 NOS specific inhibitor 1400W (n = 3–4, * = p < 0.01). (**C**) Female AMs from *Nos3*$^{-/-}$ mice lose the in vitro killing advantage of wild-type female AMs and show the same killing rate as wild-type or NOS3 deficient male AMs (n = 3, * = p < 0.01 vs wild-type). (**D**) In vivo, absence of NOS3 reduces, but does not completely eliminate, the female advantage in bacterial clearance (n = 15, * = p<0.015 vs all 3 other groups) and results in increased mortality from pneumococcal pneumonia (**E**) (n = 12 female mice per group, * = p < 0.01). Conversely, transgenic male mice with increased expression of human NOS3 show enhanced killing of *S. pneumoniae* in vivo (**F**) (lower bacterial survival, n ≥ 5, * = p < 0.01). In this low-dose inoculum model, NOS2 deletion (**G**) or inhibition (**H**) causes reduced bacterial clearance in male, but not female mice (n = 8, * = p < 0.05).

another NOS isoform, namely the constitutively expressed endothelial type NOS3 found in alveolar and other macrophages (*Miles et al., 1998*; *van Straaten et al., 1998*; *Connelly et al., 2003*). This postulate was directly tested using mice genetically deficient in NOS3. In vitro, deficiency in NOS3 did not alter the killing by male macrophages, but completely abolished the female advantage in bacterial killing (*Figure 3C*). In vivo, absence of NOS3 had minimal effects on male clearance of bacteria, but reduced the female advantage by ~50% (*Figure 3D*). Genetic deletion of NOS3 greatly reduced survival in female mice with pneumonia (*Figure 3E*), and transgenic overexpression of NOS3 increased bacterial clearance in male mice (*Figure 3F*), supporting the functional importance of this pathway. The 24 hr in vivo bacterial clearance assay provides enough time for induction of NOS2, which is known to mediate macrophage antimicrobial function in rodents (*Nathan and Shiloh, 2000*). To test the contribution of this isoform, we measured bacterial clearance in NOS2-deficient mice and in mice treated with the NOS2 inhibitor 1400W. The data show that male mice do rely on NOS2, since clearance of pneumococci was markedly worse in NOS2-deficient or 1400W-treated mice compared to controls (*Figure 3G,H*). In contrast, female mice showed a trend for improved clearance when *Nos2* was genetically deleted (*Figure 3G*); similar results were seen in wild-type female mice after pharmacologic inhibition of NOS2 (*Figure 3H*). Since NO itself can inhibit NOS3 function (*Buga et al., 1993*; *Erwin et al., 2005*), the data suggest that low-level basal NOS2 or NOS2 induction actually hampers the beneficial activity of NOS3 in females.

## Estrogen effects on NOS3

To facilitate biochemical analysis of estrogen effects on macrophage function, we tested the effect of estrogen on bacterial uptake and killing by macrophage cell lines. The cell lines were cultured using hormone-free serum. After estrogen treatment, the macrophage cell lines J744A.1 (murine) and U937 (human) showed increased killing of internalized bacteria (*Figure 4A,B*), although no differences in binding or internalization of bacteria were observed (data not shown). This increased killing capacity was inhibited by the NOS inhibitors NLA or L-NMMA, but not their stereoisomer controls. After estrogen treatment (0.2 ng/ml, 1 hr), we measured slight, albeit consistent, increases in nitrite production by bacteria-challenged J774A.1 macrophages (p < 0.01, paired *t*-test, n = 13). The increase (~25 pmoles $NO_2^-$ per $10^6$ macrophages) is consistent with the small levels attributed to NOS3 activity in normal rat AMs (~70 pmoles/$10^6$ cells (*Miles et al., 1998*)) and with the rate of NO release observed upon estrogen stimulation of human monocytes (*Stefano et al., 1999*).

Studies of NOS3 in endothelial cell biology have identified rapid activation via phosphorylation of NOS3 at serine 1177 (*Gonzalez et al., 2002*) and have shown that estrogen mediates this activation via specific signaling cascades, such as phospho-inositol-3 kinase and protein kinase B, also known as Akt, (*Chambliss and Shaul, 2002*; *Duckles and Miller, 2010*). To investigate the pathway in macrophages, we cultured J774A.1 macrophages in hormone-free serum. Western blot analysis of responses after addition of estrogen to J774A.1 cells showed rapid phosphorylation of Akt and NOS3 in response to estrogen (*Figure 4C*; fold-increase 1.7 $\pm$ 0.3, 1.7 $\pm$ 0.1, respectively, n = 4–5, p < 0.01) and basally increased levels of pNOS3 and pAkt in normal female mouse AMs compared to male AMs (*Figure 4C*, fold-increase 1.7 $\pm$ 0.4, 1.5 $\pm$ 0.2, pAkt and pNOS3 respectively, n = 2–3, p < 0.01). The Western blot analysis also shows that the amount of NOS3 in the macrophage cell line and primary AMs is quite low, at least two orders of magnitude lower than present in the positive control bEnd.1 endothelial cell line samples. A role for PI3K in the signaling cascade was supported by the decrease in pAkt levels, either basally or after E2-treatment, in J774A.1 macrophages treated with the inhibitor wortmannin (*Figure 4D*; fold increase 2.3 $\pm$ 0.2, 1.04 $\pm$ 0.5, pAkt after E2 or E2 + wortmannin, respectively, n = 2–3, p < 0.05).

To test this pathway more directly, we also used the Akt inhibitor, 1L-6-hydroxymethyl-chiro-inositol 2-(R)-2-O-methyl-3-O-octadecylcarbonate (*Takeuchi and Ito, 2004*), to test whether it would block macrophage killing of pneumococci. This agent did not alter bacterial binding or internalization (not shown). As illustrated in *Figure 4E*, Akt inhibition completely abrogated estrogen-mediated enhanced killing, supporting a functional role for Akt in estrogen-mediated activation of macrophages.

To begin testing of the translational potential of targeting NOS3, we first tested whether local delivery of estrogen to the lungs would improve resistance to bacterial pneumonia. We found that aerosol treatment of male mice before challenge with pneumococci improved the clearance of bacteria (*Figure 4F*). This was especially true if we used an estradiol conjugated to albumin, which acts to slow absorption of this lipophilic hormone (systemic uptake being the basis for a form of aerosolized estrogen-replacement therapy formerly used in menopausal women [*Studd et al., 1999*]). Populations

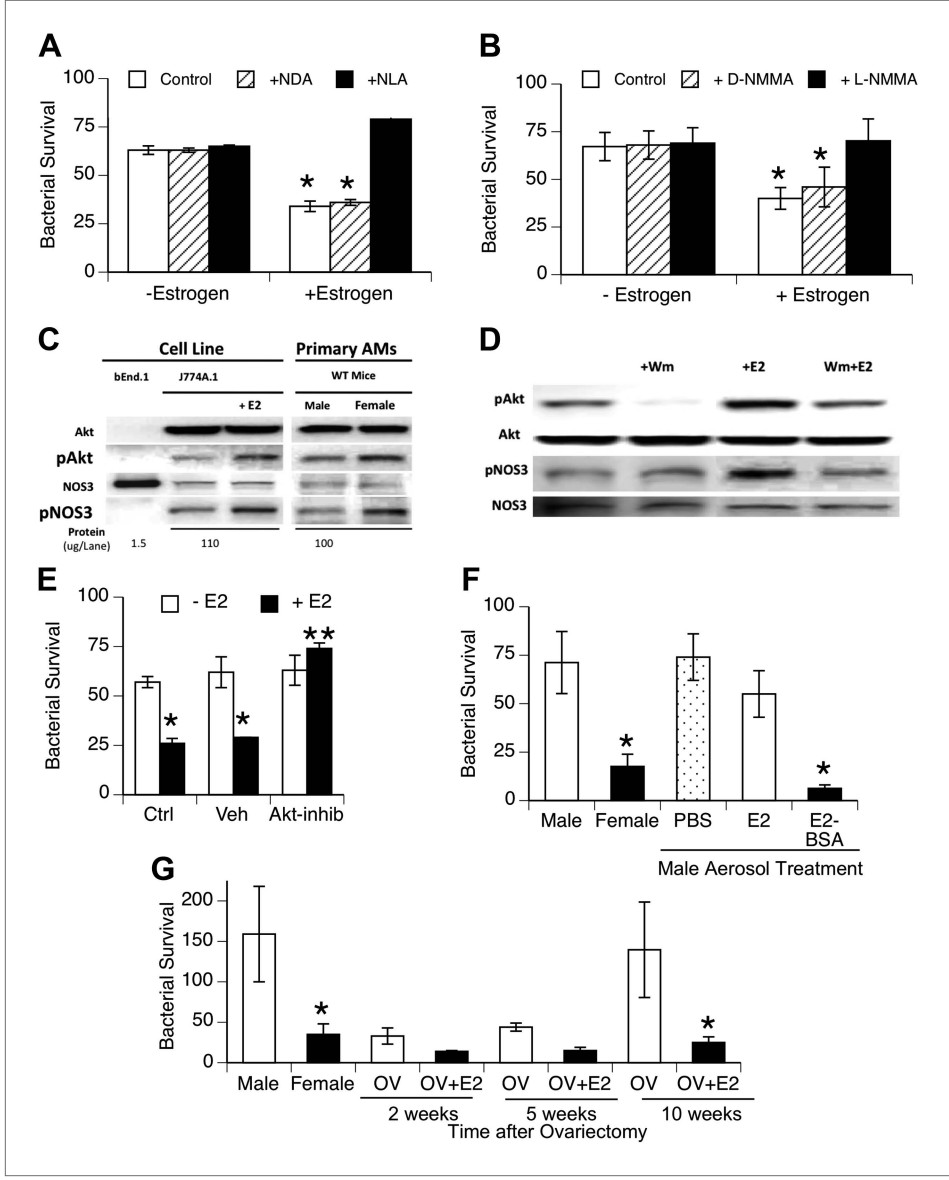

**Figure 4**. Estrogen-mediated activation of macrophage NOS3. Estrogen treatment of J774A.1 mouse or human U937 macrophages (**A** and **B**) increases killing of ingested pneumococci; this increased killing is prevented by the NOS inhibitors NLA or l-NMMA, but not control stereoisomers (n = 3–4, * = p < 0.01). (**C**) Western blot analysis shows >100-fold NOS3 in macrophages compared to the endothelial cell line bEnd.1; after 30 min, estrogen-treated (E2, estradiol, 0.2 ng/ml) J774A.1 mouse macrophages show increased phosphorylation of Akt and NOS 3, while normal female AMs show basally increased pAkt and pNOS3 compared to male AMs; (**D**) basal- and estrogen-enhanced phosphorylation of Akt and NOS3 are inhibited by wortmannin (Wm, 50 nM). (**E**) Inhibition of Akt with 1L-6-hydroxymethyl-chiro-inositol 2-(R)-2-O-methyl-3-O-octadecylcarbonate (10 μg/ml REF) prevents estrogen-mediated increased bacterial killing in J774A.1 cells (n = 3, * = p < 0.01). (**F**) Aerosol pre-treatment of male mice with albumin-conjugated estrogen 30 min before pneumococcal infection improves bacterial killing (n = 6, * = p < 0.01). (**G**) In ovariectomy-model of menopause, female mice lose their greater resistance to pneumococcal pneumonia after 10 weeks, an effect reversed by treatment with estrogen prior to infection, n ≥ 8 for control, 10 week groups; n = 3 for 2 and 5 week groups; * = p < 0.01.

at risk for localized outbreaks of primary pneumococcal pneumonia include elderly (post-menopausal) nursing home residents (*Muder, 1998*). We evaluated bacterial clearance in a mouse model of menopause that gradually develops several weeks after surgical ovariectomy (*Chakraborty and Gore, 2004*). We found a gradual decline in bacterial clearance in ovariectomized female mice to normal male levels at

10 weeks after ovariectomy; aerosolized estrogen treatment of ovariectomized mice reversed this decline and restored the normal, superior female clearance capacity (*Figure 4G*).

## Targeting NOS3 in primary pneumococcal pneumonia

Statins can increase levels of NOS3 (*Forstermann and Li, 2011*) and have been associated with beneficial effects on incidence of hospitalization for pneumonia and subsequent mortality (*Thomsen et al., 2008*; *Nielsen et al., 2012*). To study possible effects on innate immunity and initial resistance to pneumonia, we measured the effects of statin therapy and estrogen-replacement therapy on the incidence of hospitalization with pneumonia in a large well-defined female human population in Denmark between 1997 and 2012. *Table 1* shows the characteristics of 28,576 female subjects with first-time hospitalized pneumonia and 142,880 age-matched female population control subjects. The unadjusted OR results in *Table 2* indicate that in crude analyses, estrogen users had a similar pneumonia hospitalization risk than estrogen non-users, and statin users had a slightly higher pneumonia risk than statin non-users. After controlling for comorbidity and other confounding factors associated with drug use, we found that receiving statin therapy reduced incidence of pneumonia requiring hospitalization in females. In addition, estrogen-replacement therapy was associated with a similar reduction, and women receiving both statin and estrogen therapy showed an even greater reduction in pneumonia risk (adjusted OR 0.67, 95% CI: 0.60–0.75).

To investigate statin effects on macrophage interaction with bacteria, we first found that in vitro treatment of macrophages for 1–3 hr with mevastatin led to increased levels of NOS3 and its phosphorylated isoform (*Figure 5A*; fold-increase 2.0 $\pm$ 0.2, 1.7 $\pm$ 0.6, pNOS3, NOS3, 3 hr after mevastatin, respectively, n = 2–3, p < 0.05). This was associated with killing of internalized bacteria (*Figure 5B*). We then measured in vivo clearance of pneumococci in mice treated with statins and found improved clearance

**Table 1.** Characteristics of 28,576 female subjects with first-time hospitalized pneumonia and 142,880 age-matched female population control subjects from Northern Denmark, 1997–2012

| Characteristic at time of pneumonia admission | Pneumonia cases | Population controls | Total |
|---|---|---|---|
| n | 28,576 | 142,880 | 171,456 |
| **Age** | | | |
| 15–39 years | 2006 (7.0) | 10,062 (7.0) | 12,068 (7.0) |
| 40–64 years | 6383 (22.3) | 32,023 (22.4) | 38,406 (22.4) |
| 65–79 years | 9871 (34.5) | 49,267 (34.5) | 59,138 (34.5) |
| ≥80 years | 10,316 (36.1) | 51,528 (36.1) | 61,844 (36.1) |
| Charlson comorbidity index | | | |
| Index low (0) | 12,031 (42.1) | 100,297 (70.2) | 112,328 (65.5) |
| Index medium (*Mizgerd, 2006*; *Shrestha et al., 2013*) | 11,324 (39.6) | 34,896 (24.4) | 46,220 (27.0) |
| Index high (≥3) | 5221 (18.3) | 7687 (5.4) | 12,908 (7.5) |
| Alcoholism-related conditions | 1398 (4.9) | 2225 (1.6) | 3623 (2.1) |
| Preadmission medication use | | | |
| Antibiotic use (≤3 months) | 12,778 (44.7) | 20,002 (14.0) | 32,780 (19.1) |
| Statins, current use (≤ 6 months) | 3506 (12.3) | 16,111 (11.3) | 19,617 (11.4) |
| Estrogen, current use (≤ 6 months) | 2592 (9.1) | 13,225 (9.3) | 15,817 (9.2) |
| Statins and estrogen, current use (≤ 6 months) | 509 (1.8) | 2509 (1.8) | 3018 (1.8) |
| Statins and estrogen, no current use | 21,969 (76.9) | 111,035 (77.7) | 133,004 (77.6) |
| Marital status | | | |
| Married | 10,109 (35.4) | 57,673 (40.4) | 67,782 (39.5) |
| Never married | 2825 (9.9) | 13,523 (9.5) | 16,348 (9.5) |
| Divorced or widowed | 15,642 (54.7) | 71,684 (50.2) | 87,326 (50.9) |

**Table 2.** Odd ratios for first-time hospitalized pneumonia associated with current use of statins alone, estrogen alone, and estrogen + statins in combination

| Statin use | Pneumonia cases (n = 28,576) | Matched population controls (n = 142,880) | Crude OR (95% CI)* | Adjusted OR (95% CI)† |
|---|---|---|---|---|
| No current use of statins or estrogen | 21,969 (76.9) | 111,035 (77.7) | 1.0 (reference) | 1.0 (reference) |
| Current use of statins (≤ 180 days before admission) | 3506 (12.3) | 16,111 (11.3) | 1.12 (1.07–1.16) | 0.82 (0.78–0.85) |
| Current use of estrogen (≤ 180 days before admission) | 2592 (9.1) | 13,225 (9.3) | 0.99 (0.95–1.04) | 0.82 (0.78–0.86) |
| Current use of both statins and estrogen (≤ 180 days before admission) | 509 (1.8) | 2509 (1.8) | 1.04 (0.94–1.15) | 0.67 (0.60–0.75) |

*Matched for age and hospitalization date.

†Adjusted for level of Charlson's comorbidity index (19 different comorbidities), alcoholism-related conditions, antibiotics before admission, and marital status (see **Table 1**).

in both genders, but this improvement was not seen when mice deficient in NOS3 were used (**Figure 5C**). Male mice treated with statins showed improved survival with pneumococcal pneumonia (**Figure 5D**).

AVE3085 is a small-molecule compound that increases NOS3 mRNA and protein levels by mechanisms distinct from statins (e.g., by enhancing transcription) and may provide a more NOS3-specific therapeutic option (**Wohlfart et al., 2008**). We found that AVE3085 treatment improved bacterial killing by macrophages in vitro (**Figure 5E**) and that mice treated with AVE3085 either by oral or subcutaneous administration showed substantially increased clearance of pneumococci in vivo, an effect not observed in Nos3−/− mice (**Figure 5F**).

## Targeting NOS3 in secondary pneumococcal pneumonia

Secondary pneumococcal pneumonia remains a major problem after primary influenza (**Shrestha et al., 2013**) and can be modeled by infecting mice with a non-lethal dose of influenza and then challenging the lungs with pneumococci 7 days later. This is a time period of maximal susceptibility to secondary infection (**Sun and Metzger, 2008**; **Ghoneim and McCullers, 2013**), as illustrated by the remarkably low dose of pneumococci needed to cause pneumonia (500 CFU) compared to the 1,00,000 CFU used in the primary pneumonia model. We evaluated the pharmacologic agents that proved effective in our primary pneumonia model for potential benefit to reduce susceptibility or severity of secondary pneumonia after influenza. We found that treatment of mice with statins substantially improved survival in a dose-dependent manner (**Figure 6A**). We observed similar improved survival in mice treated with AVE3085 (**Figure 6B**) and also found this treatment improved bacterial clearance measured at 24 hr after challenge (**Figure 6C**).

## Discussion

We sought to identify new approaches to enhancing innate immunity to bacterial pneumonia by investigating the basis for gender differences in resistance to pneumococcal pneumonia. We conclude that estrogen mediates greater host resistance to pneumonia in female mice via effects on the constitutively expressed NOS3 in lung macrophages. The basis for this conclusion includes data in mouse cells or models using pharmacologic agents, inhibitors, and genetically deficient animals. A similar advantage is seen in female human primary AMs and estrogen-treated human macrophages. These observations identify a novel host defense mechanism and a function for the main NOS isoform found in normal human macrophages. Pharmacologic targeting of NOS3 with one type of drug already in clinical use (statins) and another small-molecule lead compound (AVE3085) led to improved bacterial clearance and improved survival from secondary pneumococcal pneumonia.

The mechanism(s) by which macrophage NOS3 contributes to improved killing of ingested bacteria include direct bactericidal mechanisms, either alone or in combination with superoxide (**Fang, 2004**),

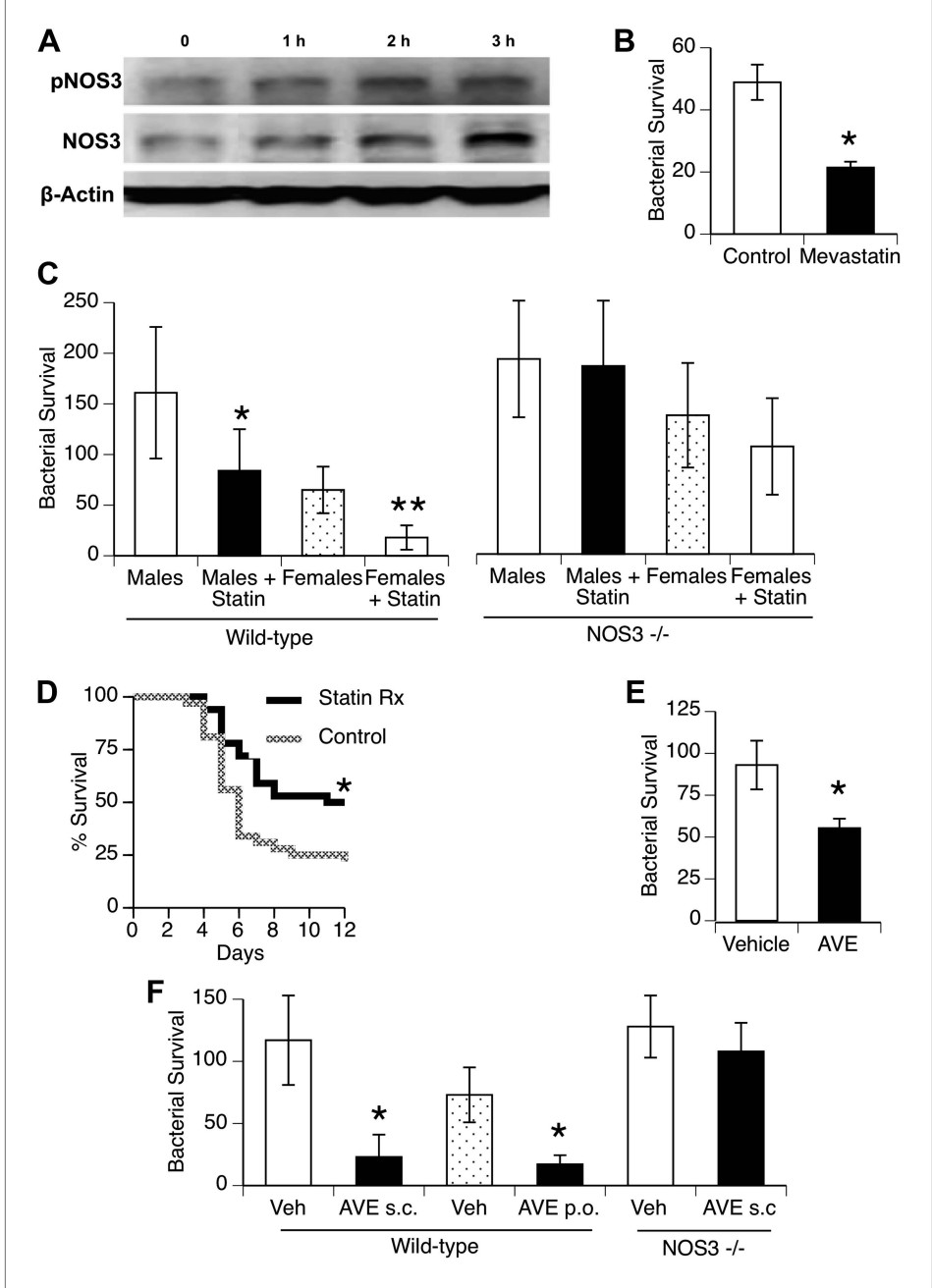

**Figure 5**. Statins enhance innate immune resistance to *S. pneumoniae* via NOS3. (**A**) In vitro treatment of J774A.1 mouse macrophages with mevastatin (5 µM) increases levels of pNOS3 and NOS3 and (**B**) concomitantly increases killing of internalized bacteria (n = 4, * = p < 0.01). (**C**) In vivo, pre-treatment of mice with pravastatin (50 mg/kg) significantly improves bacterial clearance in wild-type mice (n = 8, * = p < 0.01 vs male controls; ** = p < 0.01 vs males, males + statin), but has no significant effect on either male or female NOS3$^{-/-}$ mice. (**D**) Statin-treated male mice with pneumococcal pneumonia show improved survival (n = 8, * = p < 0.01). (**E**) AVE3085, a small molecule activator of NOS3, increases bacterial killing by mouse macrophages in vitro (n = 3, * = p < 0.01) (**F**) Pre-treatment of male mice with AVE3085 by either subcutaneous or oral route improves in vivo bacterial clearance, an effect not seen in NOS3$^{-/-}$ male mice (n = 3–8, * = p < 0.01).

but NOS3 might also act to mediate salutary intracellular signaling events (*Hernansanz-Agustín et al., 2013*). The very low amounts of NO generated by macrophage NOS3 prompt the speculation that additional mechanisms act to amplify its contribution to bacterial killing. These may include increased

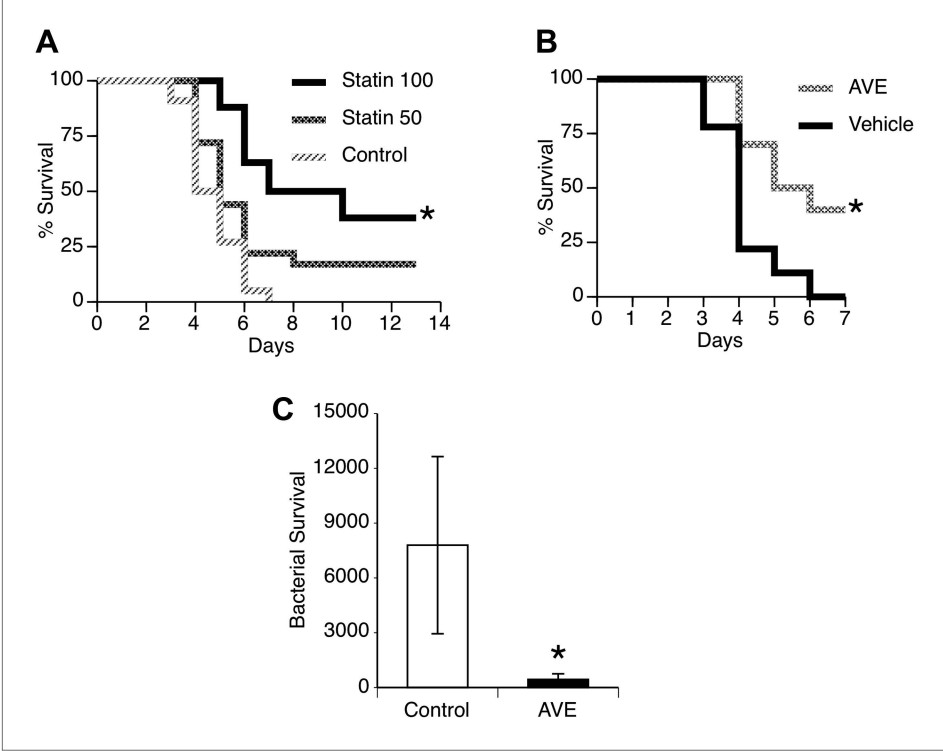

**Figure 6**. Statins and AVE3085 improve survival from post-influenza secondary pneumococcal pneumonia. Male mice were allowed to recover 7 days from mild influenza (PR8 1 PFU i.n.) and then challenged with *S. pneumoniae* (500 CFU i.n.). Pre-treatment with (**A**) pravastatin (50 or 100 mg/kg) or (**B**) AVE3085 (0.75 mg, s.c.) caused a significant improvement in survival (n = 10, * = p < 0.01). (**C**) AVE3085 treatment also lead to improved bacterial clearance 24 hr after pneumococcal challenge in this post-influenza model (n = 6, * = p < 0.01).

trafficking of NOS3 after estrogen-mediated phosphorylation to phagosomes, selective triggering of calcium-dependent NOS3 production of NO around bacteria by phagocytosis-associated calcium fluxes, and combination of NO with superoxide to generate peroxynitrite, a potent microbicidal agent (*Darrah et al., 2000*). These possibilities are not mutually exclusive. The data also show an interesting but unresolved finding of incomplete concordance between in vitro and in vivo results using NOS3-deficient mice. Specifically, NOS3 deficiency eliminated the female advantage entirely in studies of bacterial killing by AMs in vitro. In contrast, the female advantage in vivo was diminished by approximately half in NOS3-deficient mice. The mechanism(s) and significance of this persistent component of gender difference in mice are unknown and need further study.

Another aspect of our study that merits discussion is our use of a low-dose inoculum for the primary pneumonia model. This is intended to model the initial phase of pneumonia that follows the most likely route: aspiration of nasopharyngeal pneumococci or other bacteria. However, the quantity and composition of bacteria that enter the lungs during nocturnal aspiration is unknown. Moreover, while the premise that this is how pneumonia starts in humans is logical, there exists no formal proof. Ultimately, whether this attempt at a pre-clinical model proves relevant will depend on how the results translate to the clinical setting.

The potential relevance of these findings to human biology is indeed supported by the epidemiologic findings that both statins and estrogen-replacement therapy reduced risk of lung infection in a large cohort of women. There are also interesting reports of association of NOS3 polymorphisms with risk of community-acquired pneumonia (*Salnikova et al., 2013*). However, we consider that the translational potential of these findings may be most relevant for the problem of secondary pneumonia, specifically the increased susceptibility to bacterial lung infections that follows influenza. Using pre-treatment before the bacterial challenge is arguably a limitation of our studies in the primary pneumonia model; this approach is useful for proof-of-principle but does not mirror likely clinical usage. In

contrast, pre-treatment in the secondary pneumonia model does represent a realistic scenario for short-term, prophylactic immunomodulatory therapy. This approach could boost resistance to prevent bacterial pneumonias during intervals of increased susceptibility. This could well benefit patients who are identified to be at high risk of secondary pneumonia, example hospitalized individuals with severe seasonal or pandemic influenza.

## Materials and methods

### Study design

This study began by seeking to characterize the basis for greater female resistance to pneumococcal pneumonia. A murine model of pneumococcal pneumonia was used to compare male and female bacterial clearance, lung inflammation, and survival in vivo. In vitro assays of macrophage antibacterial function were used to identify killing of internalized bacteria as a critical difference. Pharmacologic inhibitors, genetic mouse models, and in vitro macrophage studies were used to investigate the role of oxidant-generating enzyme systems and estrogen. After identification of NOS3 as an important mediator, we tested pharmacologic agents with translational potential for ability to improve outcomes in primary and post-influenza secondary pneumococcal pneumonia. For in vivo studies, we used male and female mice of the same age, and the studies were not blinded. Sample sizes for all of the experiments were sufficient to detect statistically significant differences between treatment groups, with all measurements included in analyses. To explore whether the effects of estrogen and statins that were observed in the murine models also occur in people, we studied incidence of hospitalized pneumonia associated with the use of estrogen and statins in a population-based case-control study based on medical databases in Denmark. Cases of pneumonia were identified as all women who received a first-time principal hospital diagnosis of pneumonia in the former North Jutland and Aarhus Counties, Northern Denmark (1.2 million inhabitants) between 1997 and 2012. Using the Danish Civil Registration System, each case subject was matched with five population control subjects with same age, female gender, and residence in Northern Denmark on the pneumonia index date. We ascertained use of estrogen and statins in all individuals from population-based prescription databases. To control for confounding by other conditions potentially associated with both estrogen and statin use and pneumonia risk, we retrieved individual-level data on 19 major comorbid disease categories as evidenced in the Charlson comorbidity index, as well as on alcoholism-related conditions, recent antibiotic use, and marital status as a marker of socioeconomic status. We then computed odds ratios (ORs) for a first-time pneumonia admission among women with and without estrogen and statin use, using conditional logistic regression analysis to control for confounders.

### Mouse models of pneumococcal pneumonia

Normal eight- to 12-week old male and female mice C57BL/6 mice from Charles River Laboratories (Wilmington, MA) or Jackson Laboratories (Bar Harbor, ME) were used. C57BL/6 mice genetically deficient in nitric oxide synthase 2 (*Laubach et al., 1995*), nitric oxide synthase 3 (*Shesely et al., 1996*), and NADPH oxidase (*Morgenstern et al., 1997*) were obtained from Jackson Laboratories (Bar Harbor, ME). Mice transgenic for human NOS3 were previously characterized (*Jones et al., 2003*). All animals were housed in sterile microisolator cages in a barrier facility and had no evidence of spontaneous infection. Prior approval for all experimentation was obtained from the institutional animal use review committee.

Primary pneumococcal pneumonia was modeled as previously reported (*Arredouani et al., 2004*). Pneumonia was induced by intranasal instillation (i.n.) of 25 μl of a bacterial suspension containing approximately $10^5$ colony-forming units (CFU) of *S. pneumoniae* type 3 of mice under short-term anesthesia with inhaled isoflurane. The bacterial suspension was prepared to contain $10^5$ CFU based on optical density calculations. However, results of direct validation of actual CFU in these samples were always performed and showed actual CFU values that ranged from 90–110 % of the calculated $10^5$ CFU value. Hence, the dose delivered is stated as ~$10^5$ CFU. For analysis at 4 or 24 hr post-infection, mice were sacrificed with/by lethal overdose of intraperitoneal sodium pentobarbital (FatalPlus, Vortech Pharmaceuticals, Dearborn, MI) or by excess inhaled isoflurane. To measure total lung bacteria counts (CFU), whole lungs were harvested and homogenized in 1 ml sterile water with a tissue homogenizer (Omni International, Warrenton, VA). Serial 10-fold dilutions in sterile water were made from these homogenates, and 100 μl volumes were plated onto sheep-blood agar plates and incubated at

37°C. CFUs were counted after 18–20 hr. In experiments to assess survival, mice were instilled i.n. with a single 25 µl suspension containing a higher dose of *S. pneumoniae* (~3 × 10⁵ CFU) and survival followed as reported in 'Results'. For analysis of lung inflammation, bronchoalveolar lavage was performed in situ with a 22-gauge catheter inserted into the proximal trachea, flushing the lower airways six times with 0.7 ml of phosphate-buffered saline (PBS). The fluid retrieved from the first lavage was kept for ELISA assays. The BAL cells were separated from the BAL fluid by centrifugation, resuspended in PBS and counted. A fraction was cytospun on microscopic slides for staining with Diff-Quick (Baxter Scientific Products, McGaw Park, IL) for subsequent differential counts. In initial experiments, male mice were analyzed 20 days after subcutaneous implantation according to manufacturer's instructions of slow-release 17-beta-estradiol pellets (~70 µg/day, Innovative Research, Sarasota, FL). In later experiments, mice received a single dose of estradiol by inhalation exposure for 1 hr to an aerosol generated from a solution of E2-BSA β-Estradiol 6-(O-carboxymethyl)oxime: BSA (75 ng/ml aerosol solution, ~30 mol steroid per mol BSA, Sigma) using the mouse exposure system described in *Hamada et al., 2003*. Drug treatments to assess effects on 24 hr bacterial clearance were administered 4 hr before and 8 and 20 hr after bacterial challenge included i.p. pravastatin 50–100 mg/kg, and 1400W 10 mg/kg. The NOS3 activator AVE3085 (2,2-difluoro-benzo[1,3]dioxole-5-carboxylic acid indan-2-ylamide, CAS no. 450348-85-3; empirical formula $C_{17}H_{13}F_2NO_3$; (*Wohlfart et al., 2008*)) was administered by gavage at 100 mg/kg/day for 3 days prior to the challenge or by subcutaneous injection of 0.75 mg 24 and 3 hr before the challenge.

To model post-influenza secondary pneumonia, mice were instilled i.n. on day 0 with a single 25 µl or 50 µl suspension containing influenza virus (A/PR 8/34; H1N1), under general anesthesia by i.p. ketamine (120 mg/kg) plus xylazine (16 mg/kg). Influenza-treated animals routinely lost weight and then recovered by day 7, when the secondary infection was administered. On day 7 after initial influenza infection, mice were subjected to the same anesthesia and then instilled i.n. with a single 25 µl of bacterial suspension containing approximately 500 CFU of *S. pneumoniae* type 3. Pravastatin (50–100 mg/kg i.p.) or AVE3085 (0.75 mg s.c.) was administered daily starting one day before the pneumococcal challenge. Subsequent analyses were performed as for the primary pneumonia model described above.

## Cell isolation and culture

After euthanasia, mouse AMs obtained by repeated lung lavage with sterile PBS were centrifuged at 150×*g* and resuspended in HBSS⁺ (Hanks' Buffered Salt Solution with calcium, 0.3 mM, and magnesium, 1 mM). Primary mouse AMs were used immediately without exposure to culture media components that contain estrogen (e.g. serum) or (weakly) estrogenic phenol red. The mouse and human macrophage cell lines J774A.1 and U937 were obtained from ATCC and maintained in phenol-free RPMI with 10% FBS. In experiments testing effects of estrogen addition in vitro, a charcoal-stripped FBS (Hyclone, Logan, UT) was used for macrophage culture to eliminate exposure to serum hormones. Human alveolar macrophages were obtained from non-smoking volunteers by bronchoalveolar lavage and resuspended in HBSS⁺ when tested immediately or cultured in phenol-free RPMI with 10% fetal bovine serum (FBS) overnight. The potential confounding effects of estrogen in FBS were not observed in human AM samples cultured overnight, that is gender differences observed in cells used immediately after isolation were preserved. All human subject experimentation was conducted under approved protocols reviewed by the institutional review boards. Estrogen was added to macrophage cultures as β-Estradiol-Water Soluble (Sigma) at 0.05–0.3 ng/ml. All other reagents not otherwise specified were obtained from Sigma Chemical, St. Louis, MO.

## Bacteria

*S. pneumoniae* serotype 3, *E. Coli*, *S. aureus* (#6303, 19138, 25923, respectively, ATCC, Rockville, MD) were cultured overnight on 5% sheep blood-supplemented agar Petri dishes (VWR # 90001-282, West Chester, PA). A stock suspension was prepared and aliquots kept at −80°C. For each experiment, an aliquot was grown overnight on a blood agar plate and resuspended in sterile PBS. Bacterial concentration of the obtained suspension was estimated by OD₆₀₀ measurements, comparing to a prior standard curve of OD₆₀₀ vs CFU. The appropriate dilution was prepared in sterile PBS to be administered to mice, and this estimate was checked in parallel by CFU assay to determine the precise concentration.

## Phagocytosis and bacterial killing assay

Macrophage binding, internalization, and killing of internalized bacteria were measured using CFU assays of cell samples lysed by 10-fold excess H₂O (pH 10, 3 min). Macrophages (1.5 × 10⁶ cells/1.5 ml

HBSS⁺) were mixed with bacteria ($15 \times 10^6$ CFU: 10 bacteria per 1 macrophage) for 1 hr at 37°C. After centrifugation the cell pellet was re-suspended and an aliquot taken to measure the total cell-associated CFU (bound and internalized). After brief incubation of the cell suspension with genta- micin (200 µg/ml, 15 min at 37°C) to kill external, bound bacteria, the macrophages were washed and an aliquot taken for lysis and CFU assay to quantitate the number of internalized, live bacteria. The macrophages were then incubated an additional hour to allow killing of internalized bacteria and ali- quots taken again for CFU quantitation. The CFU data obtained at various time points allow calcula- tion of the number of bacteria bound, internalized, and killed. Effects of pharmacologic inhibitors or hormones were tested by including agents or their vehicles in the assay buffers. Respiratory burst function in normal mouse AMs was measured by quantitation of the H2O2-catalyzed oxidation of Amplex Red (Molecular Probes, Eugene, OR) to a fluorescent product after stimulation with adsorbed antibodies (anti-FcR, CD18) or PMA (100 nM) as previously described (*Józefowski and Kobzik, 2004*).

## Western blot and ELISA analysis

Western blot analysis was performed on macrophage lysates lysis buffer 1% NP-40 with protease and phosphatase inhibitors using the protocols described in *Gonzalez et al., 2002*. The endothelial cell line bEnd.3 (ATCC CRL-2299) was used as a positive control for ('endothelial') NOS3. Antibodies for NOS3, Akt, and phosphorylation-state specific isoforms used include: NOS3 rabbit pAb (C-20, Santa Cruz Biotechnology) and phospho-eNOS (Ser1177, C9C3) rabbit mAb, phospho-Akt (Ser473, D9E) rabbit mAb, Akt rabbit pAb, and Beta-Actin (13E5) rabbit mAb, all from Cell Signaling Technologies. After incubation with peroxidase-conjugated goat anti-rabbit IgG or goat anti-mouse IgG (Pierce), labeling was detected using chemiluminescence. Quantitation of digitized signal intensity data was performed using ImageJ software (http://imagej.nih.gov). Cytokines were quantitated in BAL fluid using commercially available ELISA kits (R&D Systems Inc., Minneapolis, MN) following the manufac- turer's instructions.

## Statistical analysis

Data were analyzed using the Prism software package (GraphPad Software). Differences in Kaplan- Meier survival curves were analyzed using a log-rank test. For most other measurements, differences between groups were examined by ANOVA with Tukey's multiple comparison test. For gel quantita- tion values, the visual impression of increased intensity was analyzed using one-way t-tests. Data are presented as mean ± SD. Results were considered significant when p value was <0.05.

## Acknowledgements

We thank J Stamler, T Michel, J Mizgerd, S Matalon, and E Kovacs for helpful discussion and advice and J Soukup for technical assistance. Funding: this research was supported by NIH HL083436, ES0002, and the Flight Attendant Medical Research Institute.

## Additional information

### Competing interests

HR: Named as inventor on patents (US 7,179,839; 8,309,608, held by Sanofi, Inc.) describing AVE3085 as a useful compound for cardiovascular indications. PW: Named as inventor on patents (US 7,179,839; 8,309,608, held by Sanofi, Inc.) describing AVE3085 as a useful compound for cardiovascular indications. The other authors declare that no competing interests exist.

### Funding

| Funder | Grant reference number | Author |
| --- | --- | --- |
| National Heart, Lung, and Blood Institute | HL083436 | Lester Kobzik |
| National Institute of Environmental Health Sciences | ES0002 | Lester Kobzik |
| Flight Attendant Medical Research Institute | | Lester Kobzik |

The funders had no role in study design, data collection and interpretation, or the decision to submit the work for publication.

## Author contributions

ZY, JAK, HTS, SJ, AC, Acquisition of data, Analysis and interpretation of data, Drafting or revising the article; Y-CTH, HK, Acquisition of data, Drafting or revising the article, Contributed unpublished essential data or reagents; RC, HR, PW, Drafting or revising the article, Contributed unpublished essential data or reagents; RWT, Acquisition of data, Analysis and interpretation of data, Drafting or revising the article, Contributed unpublished essential data or reagents; LK, Conception and design, Analysis and interpretation of data, Drafting or revising the article

## Ethics

Human subjects: We studied incidence of hospitalized pneumonia associated with use of estrogen and statins in a population-based case-control study based on medical databases in Denmark. Cases of pneumonia were identified as all women who received a first-time principal hospital diagnosis of pneumonia in the former North Jutland and Aarhus Counties, Northern Denmark (1.2 million inhabitants) between 1997 and 2012. Using the Danish Civil Registration System, each case subject was matched with five population control subjects with same age, female gender, and residence in Northern Denmark on the pneumonia index date. The study was approved by the Danish Data Protection Agency (record number: 2013-41-1924). Danish registry data are generally available for research purposes, and, according to Danish law, use of the data does not require informed consent.

Animal experimentation: This study was performed in strict accordance with the recommendations in the Guide for the Care and Use of Laboratory Animals of the National Institutes of Health. All of the animals were handled according to approved institutional animal care and use committee (Harvard Medical Area IACUC) protocol (#03287).

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
