## [Decision Letter]

Thank you for sending your work entitled “Female Resistance to Pneumonia Identifies Lung Macrophage Nitric Oxide Synthase-3 as a Therapeutic Target” for consideration at *eLife*. Your article has been favorably evaluated by Tadatsugu Taniguchi (Senior editor), a Reviewing editor, and 3 reviewers.

The Reviewing editor and the reviewers discussed their comments before we reached this decision, and the Reviewing editor has assembled the following questions and comments to help you prepare a revised submission.

We believe the question you have addressed is an important one, the experiments have been well conceived, and the results are novel.

The figures are central to the report and they need standardization (e.g. use of bold lines for treatment). The figure legends need clarification and better explanation (e.g. doses used in Figure 1, the significant differences in Figure 3). In addition:

1) Please clarify the gender of the mice for each experiment and each figure.

2) Please clarify the following: from the Materials and methods section it states under “cell isolation and culture” that macrophage cultures were incubated with estrogen. The estrogen receptor beta that is equally expressed on male and female cells and thus addition of exogenous estrogen would activate both (given macrophages from both sexes seem to express the same level of NOS3)? Is this the case or was estrogen only added to the macrophage cell line studies? If this is the case and no exogenous estrogen is added, are the authors stating that the primary macrophages obtained from males/females are wired post isolation by their exposure to estrogen whilst *in situ*?

3) What is the likely mechanism by which elevated NOS3 activation in females leads to increased bactericidal activity? Please review briefly polymorphisms within NOS3 being associated with pneumonia.

4) AVE3085 acts by increasing NOS3 transcription but the NOS3 would still need to be activated. How is this achieved in the male mice?

5) Please comment on the following; inoculating volume in which bacteria are administered can affect the phenotype/potency of the response. Perhaps utilizing the same volume may result in bacteria not reaching the same depths of the lung in females versus males because of their different size?

6) How and why are the Charlson index or other factors confounding factors in the relationship between estrogen use and pneumonia?

---

## [Author Response]

*The figures are central to the report and they need standardization (e.g. use of bold lines for treatment). The figure legends need clarification and better explanation (e.g. doses used in*
Figure 1*, the significant differences in*
Figure 3*)*.

We have reviewed the figures and tried to improve standardization, but we must confess that we are not sure which specific components or graphic element this comment is trying to identify. We will be glad to work with the editorial office to further standardize the figures, once given specific instructions. There are only two agents in Figure 1 for which ‘doses used’ applies. We are uncertain whether this request for clarification applies to one or the other or both, so we have tried to improve both. The dose of bacteria delivered to the lung is given as “∼10^5^ CFU”. We have elaborated on the reason why the CFU value is approximate by including new text in the methods section. Regarding the comment “lung samples from female mice (and estrogen-treated male mice)”, to clarify the dose of estrogen directly within the figure legend, we have modified the text to add the details also presented in the Methods section.

We have clarified the meaning of the symbol for significant differences in the figure legend for Figure 3 as follows: (D) In vivo, absence of NOS3 reduces, but does not completely eliminate, the female advantage in bacterial clearance (n = 15, * = p < .015 vs. all 3 other groups) N.B. Additional change, not requested by the review: During the review for this point about Figure 3, we noted that one of the p values is actually p = .015, rather than all of them being < .01 as originally stated. Hence for accuracy we have modified the p value here as p < .015. We consider that this accurately summarizes all the values without introducing unnecessary clutter, but will be glad to itemize each comparison if needed.

*1) Please clarify the gender of the mice for each experiment and each figure*.

We have reviewed all the figures and figure legends as requested. We have added additional information regarding mouse gender in the figure legends 3E, 5F and 6 (not copied here as they are the simple insertion of a single adjective, e. g. ‘male’). We consider that the gender is clear for the other figures and experiments, but will be glad to revise further if needed.

2) Please clarify the following: from the Materials and methods section it states under “cell isolation and culture” that macrophage cultures were incubated with estrogen. The estrogen receptor beta that is equally expressed on male and female cells and thus addition of exogenous estrogen would activate both (given macrophages from both sexes seem to express the same level of NOS3)? Is this the case or was estrogen only added to the macrophage cell line studies? If this is the case and no exogenous estrogen is added, are the authors stating that the primary macrophages obtained from males/females are wired post isolation by their exposure to estrogen whilst in situ?

We have clarified these points by first noting in the Methods that all experiments done with primary macrophages were performed immediately after lavage (for all mouse samples). We only added exogenous estrogens to the cell lines as stated in the same Methods section. To partially address the final point here, we note that for human AMs the potential effects of estrogen in FBS (that prompted the use of charcoal-stripped, hormone-free FBS in our macrophage cell line cultures) were not observed. So the primary macrophages obtained from males/females do appear to be “wired” by their in situ exposures, at least for a period of 12-24 hours. We have not addressed the duration of the effect in detail, but have noted that the effect of exogenous estrogen of the macrophage cell lines persists thru 24 hours but does decline after that (only studied at 48 hours, but this may be influenced by cell division, etc.). This is now stated more explicitly in the Methods section (“Human alveolar macrophages were obtained…”).

*3) What is the likely mechanism by which elevated NOS3 activation in females leads to increased bactericidal activity? Please review briefly polymorphisms within NOS3 being associated with pneumonia*.

For the first point, we attempted in the original submission to cover this point by a section in the Discussion that we consider to address the question sufficiently. We will be glad to elaborate further if needed, but consider that the references provided are comprehensive.

For the second point, we can find 4 articles in PubMed that study NOS3 polymorphisms and pneumonia. There are 3 English-language papers, all from the same group of investigators. They do find some association of NOS3 SNPs with community acquired pneumonia, but their populations are mostly male (e.g. >90% in the study cited below) and the associations are not significant after adjustment for multiple comparisons. Hence to address this point, and as a convenience for the interested reader, we have added this sentence and one of the three references to the Discussion: “There are also interesting reports of association of NOS3 polymorphisms with risk of community-acquired pneumonia (Salnikova, 2013).”

*4) AVE3085 acts by increasing NOS3 transcription but the NOS3 would still need to be activated*. *How is this achieved in the male mice?*

This is a very good point, and the hardest question for us to answer. Several post-translational mechanisms have been described increasing or decreasing NOS3 activity, primarily from studies of vascular endothelium (for a review see Qian and Fulton, Front Physiol; 2013, 4, 347). Although we have not investigated such posttranslational mechanisms in lung macrophages so far, we speculate that similar signals might be involved, e.g. physical signals analogous to shear stress, such as generated by macrophage movement within the alveolus or cyclic stretch of the lung during breathing, or activation signals generated during the act of phagocytosis. In addition, uptake of certain bacteria is associated with activation of the PI3K-Akt pathway (e.g. for Staph. aureus, Infect Immun. 2011 Nov;79(11):4569-77; for Shigella, Sci Signal. 2011 Sep 20;4(191):ra61). However, a short and accurate answer is that we don’t know the mechanism. This would need to be addressed by future studies.

*5) Please comment on the following; inoculating volume in which bacteria are administered can affect the phenotype/potency of the response*. *Perhaps utilizing the same volume may result in bacteria not reaching the same depths of the lung in females versus males because of their different size?*

We are very grateful for this comment because we actually tested this point early on in these studies, but neglected to report the findings. Hence, we have added this point to the first section of the Results section (“Pilot experiments compared efficiency of delivery…).

6) How and why are the Charlson index or other factors confounding factors in the relationship between estrogen use and pneumonia?

To examine the association between estrogen and statin use (exposure) and risk of pneumonia hospitalization (outcome), we aimed to control for confounding by conditions potentially associated with both estrogen and statin use and pneumonia risk. These conditions were defined by us as a priori potential confounding factors, based on previous findings in the literature, and biological considerations.

Comorbidity increases risk of pneumonia (Søgaard M, Respiratory Medicine 2014). As a measure of overall comorbidity, we computed the Charlson comorbidity index score for each subject (Charlson ME, J Chronic Dis 1987). The Charlson comorbidity index was originally validated for prediction of mortality, but has the advantage of including almost all known major risk factors for pneumonia (Thomsen RW, Diabetes Care 2004), such as chronic obstructive pulmonary disease (COPD) and other pulmonary diseases, previous cardiovascular and cerebrovascular diseases, chronic heart failure, diabetes, cancer, renal and liver disease, dementia, and HIV/AIDS (Mor A et al, Clinical Epidemiology 2013). Many of these chronic diseases are related to age, unhealthy lifestyle, and socioeconomic status (Barnett K, Lancet 2012). As other markers of health consciousness and socioeconomic status, we therefore retrieved data on alcoholism-related conditions, recent antibiotic use, and marital status, which are also known risk factors for pneumonia (Kornum JB, Eur Resp J 2012; Mor A, Clinical Epidemiology 2013). People who use preventive medications such as estrogen and statins may be more “healthy”, that is, have higher health consciousness and a healthier lifestyle and thus fewer lifestyle related diseases than other people (Brookhart A, Am J Epidemiol 2007; Shrank WH, J Gen Intern Med 2011). On the other hand, medical indications for these drugs during the study period included comorbidities that may have been already present in the patient (e.g. osteoporosis, cardiovascular disease). Since the above factors are a priori expected to be skewed both according to risk of pneumonia and chance of being drug exposed - which our analyses of 2 x 2 tables confirmed (Table S1 shows association with pneumonia case status) – we included them as potential confounders. To evaluate the actual confounding caused by the above factors, we calculated crude and adjusted odds ratios (ORs) for a first-time pneumonia admission among women with and without estrogen and statin use, while using conditional logistic regression analysis to control for the confounders: Charlson comorbidity index level (19 different comorbidities), alcoholism-related conditions, antibiotics before admission, and marital status). The changes in risk estimates, e.g. from a crude pneumonia OR in estrogen users of 0.99 (0.95-1.04) to a confounder-adjusted pneumonia OR in estrogen users of 0.82 (0.78-0.86) (Table 1), show that these factors confounded the association between estrogen use and pneumonia risk. In other words, estrogen users and non-users appeared to have similar pneumonia risk in crude analyses. When adjusting for differences in prevalence of confounding factors, estrogen users had a decreased pneumonia risk. We have added a few explanatory comments on this in the revised manuscript, Results section.